# Estimation, Optimization, and Parallelism when Data is Sparse

**John C. Duchi**[1,2]      **Michael I. Jordan**[1]
University of California, Berkeley[1]
Berkeley, CA 94720
{jduchi,jordan}@eecs.berkeley.edu

**H. Brendan McMahan**[2]
Google, Inc.[2]
Seattle, WA 98103
mcmahan@google.com

## Abstract

We study stochastic optimization problems when the *data* is sparse, which is in a sense dual to current perspectives on high-dimensional statistical learning and optimization. We highlight both the difficulties—in terms of increased sample complexity that sparse data necessitates—and the potential benefits, in terms of allowing parallelism and asynchrony in the design of algorithms. Concretely, we derive matching upper and lower bounds on the minimax rate for optimization and learning with sparse data, and we exhibit algorithms achieving these rates. We also show how leveraging sparsity leads to (still minimax optimal) parallel and asynchronous algorithms, providing experimental evidence complementing our theoretical results on several medium to large-scale learning tasks.

## 1 Introduction and problem setting

In this paper, we investigate stochastic optimization problems in which the *data* is sparse. Formally, let $\{F(\cdot; \xi), \xi \in \Xi\}$ be a collection of real-valued convex functions, each of whose domains contains the convex set $\mathcal{X} \subset \mathbb{R}^d$. For a probability distribution $P$ on $\Xi$, we consider the following optimization problem:

$$\underset{x \in \mathcal{X}}{\text{minimize}} \ f(x) := \mathbb{E}[F(x; \xi)] = \int_\Xi F(x; \xi) dP(\xi). \tag{1}$$

By data sparsity, we mean the samples $\xi$ are sparse: assuming that samples $\xi$ lie in $\mathbb{R}^d$, and defining the support $\text{supp}(x)$ of a vector $x$ to the set of indices of its non-zero components, we assume

$$\text{supp} \, \nabla F(x; \xi) \subset \text{supp} \, \xi. \tag{2}$$

The sparsity condition (2) means that $F(x; \xi)$ does not "depend" on the values of $x_j$ for indices $j$ such that $\xi_j = 0$.[1] This type of data sparsity is prevalent in statistical optimization problems and machine learning applications; in spite of its prevalence, study of such problems has been limited.

As a motivating example, consider a text classification problem: data $\xi \in \mathbb{R}^d$ represents words appearing in a document, and we wish to minimize a logistic loss $F(x; \xi) = \log(1 + \exp(\langle \xi, x \rangle))$ on the data (we encode the label implicitly with the sign of $\xi$). Such generalized linear models satisfy the sparsity condition (2), and while instances are of very high dimension, in any given instance, very few entries of $\xi$ are non-zero [8]. From a modelling perspective, it thus makes sense to allow a *dense* predictor $x$: any non-zero entry of $\xi$ is potentially relevant and important. In a sense, this is dual to the standard approaches to high-dimensional problems; one usually assumes that the data $\xi$ may be dense, but there are only a few relevant features, and thus a parsimonious model $x$ is desirous [2]. So

while such sparse data problems are prevalent—natural language processing, information retrieval, and other large data settings all have significant data sparsity—they do not appear to have attracted as much study as their high-dimensional "duals" of dense data and sparse predictors.

In this paper, we investigate algorithms and their inherent limitations for solving problem (1) under natural conditions on the data generating distribution. Recent work in the optimization and machine learning communities has shown that data sparsity can be leveraged to develop parallel (and even asynchronous [12]) optimization algorithms [13, 14], but this work does not consider the statistical effects of data sparsity. In another line of research, Duchi et al. [4] and McMahan and Streeter [9] develop "adaptive" stochastic gradient algorithms to address problems in sparse data regimes (2). These algorithms exhibit excellent practical performance and have theoretical guarantees on their convergence, but it is not clear if they are optimal—in that no algorithm can attain better statistical performance—or whether they can leverage parallel computing as in the papers [12, 14].

In this paper, we take a two-pronged approach. First, we investigate the fundamental limits of optimization and learning algorithms in sparse data regimes. In doing so, we derive lower bounds on the optimization error of *any* algorithm for problems of the form (1) with sparsity condition (2). These results have two main implications. They show that in some scenarios, learning with sparse data is quite difficult, as essentially each coordinate $j \in [d]$ can be relevant and must be optimized for. In spite of this seemingly negative result, we are also able to show that the ADAGRAD algorithms of [4, 9] are optimal, and we show examples in which their dependence on the dimension $d$ can be made exponentially better than standard gradient methods.

As the second facet of our two-pronged approach, we study how sparsity may be leveraged in parallel computing frameworks to give substantially faster algorithms that still achieve optimal sample complexity in terms of the number of samples $\xi$ used. We develop two new algorithms, asynchronous dual averaging (ASYNCDA) and asynchronous ADAGRAD (ASYNCADAGRAD), which allow asynchronous parallel solution of the problem (1) for general convex $f$ and $\mathcal{X}$. Combining insights of Niu et al.'s HOGWILD! [12] with a new analysis, we prove our algorithms achieve linear speedup in the number of processors while maintaining optimal statistical guarantees. We also give experiments on text-classification and web-advertising tasks to illustrate the benefits of the new algorithms.

## 2  Minimax rates for sparse optimization

We begin our study of sparse optimization problems by establishing their fundamental statistical and optimization-theoretic properties. To do this, we derive bounds on the minimax convergence rate of any algorithm for such problems. Formally, let $\widehat{x}$ denote any estimator for a minimizer of the objective (1). We define the optimality gap $\epsilon_N$ for the estimator $\widehat{x}$ based on $N$ samples $\xi^1, \dots, \xi^N$ from the distribution $P$ as

$$\epsilon_N(\widehat{x}, F, \mathcal{X}, P) := f(\widehat{x}) - \inf_{x \in \mathcal{X}} f(x) = \mathbb{E}_P\left[F(\widehat{x}; \xi)\right] - \inf_{x \in \mathcal{X}} \mathbb{E}_P\left[F(x; \xi)\right].$$

This quantity is a random variable, since $\widehat{x}$ is a random variable (it is a function of $\xi^1, \dots, \xi^N$). To define the minimax error, we thus take expectations of the quantity $\epsilon_N$, though we require a bit more than simply $\mathbb{E}[\epsilon_N]$. We let $\mathcal{P}$ denote a collection of probability distributions, and we consider a collection of loss functions $F$ specified by a collection $\mathcal{F}$ of convex losses $F : \mathcal{X} \times \xi \to \mathbb{R}$. We can then define the minimax error for the family of losses $\mathcal{F}$ and distributions $\mathcal{P}$ as

$$\epsilon_N^*(\mathcal{X}, \mathcal{P}, \mathcal{F}) := \inf_{\widehat{x}} \sup_{P \in \mathcal{P}} \sup_{F \in \mathcal{F}} \mathbb{E}_P[\epsilon_N(\widehat{x}(\xi^{1:N}), F, \mathcal{X}, P)], \tag{3}$$

where the infimum is taken over all possible estimators $\widehat{x}$ (an estimator is an optimization scheme, or a measurable mapping $\widehat{x} : \Xi^N \to \mathcal{X}$).

### 2.1  Minimax lower bounds

Let us now give a more precise characterization of the (natural) set of sparse optimization problems we consider to provide the lower bound. For the next proposition, we let $\mathcal{P}$ consist of distributions supported on $\Xi = \{-1, 0, 1\}^d$, and we let $p_j := P(\xi_j \neq 0)$ be the marginal probability of appearance of feature $j \in \{1, \dots, d\}$. For our class of functions, we set $\mathcal{F}$ to consist of functions $F$ satisfying the sparsity condition (2) and with the additional constraint that for $g \in \partial_x F(x; \xi)$, we have that the $j$th coordinate $|g_j| \leq M_j$ for a constant $M_j < \infty$. We obtain

**Proposition 1.** *Let the conditions of the preceding paragraph hold. Let $R$ be a constant such that $\mathcal{X} \supset [-R, R]^d$. Then*

$$\epsilon_N^*(\mathcal{X}, \mathcal{P}, \mathcal{F}) \geq \frac{1}{8} R \sum_{j=1}^d M_j \min\left\{ p_j, \frac{\sqrt{p_j}}{\sqrt{N \log 3}} \right\}.$$

We provide the proof of Proposition 1 in the supplement A.1 in the full version of the paper, providing a few remarks here. We begin by giving a corollary to Proposition 1 that follows when the data $\xi$ obeys a type of power law: let $p_0 \in [0, 1]$, and assume that $P(\xi_j \neq 0) = p_0 j^{-\alpha}$. We have

**Corollary 2.** *Let $\alpha \geq 0$. Let the conditions of Proposition 1 hold with $M_j \equiv M$ for all $j$, and assume the power law condition $P(\xi_j \neq 0) = p_0 j^{-\alpha}$ on coordinate appearance probabilities. Then*

*(1) If $d > (p_0 N)^{1/\alpha}$,*

$$\epsilon_N^*(\mathcal{X}, \mathcal{P}, \mathcal{F}) \geq \frac{MR}{8} \left[ \frac{2}{2-\alpha} \sqrt{\frac{p_0}{N}} \left( (p_0 N)^{\frac{2-\alpha}{2\alpha}} - 1 \right) + \frac{p_0}{1-\alpha} \left( d^{1-\alpha} - (p_0 N)^{\frac{1-\alpha}{\alpha}} \right) \right].$$

*(2) If $d \leq (p_0 N)^{1/\alpha}$,*

$$\epsilon_N^*(\mathcal{X}, \mathcal{P}, \mathcal{F}) \geq \frac{MR}{8} \sqrt{\frac{p_0}{N}} \left( \frac{1}{1-\alpha/2} d^{1-\frac{\alpha}{2}} - \frac{1}{1-\alpha/2} \right).$$

Expanding Corollary 2 slightly, for simplicity assume the number of samples is large enough that $d \leq (p_0 N)^{1/\alpha}$. Then we find that the lower bound on optimization error is of order

$$MR\sqrt{\frac{p_0}{N}} d^{1-\frac{\alpha}{2}} \text{ when } \alpha < 2, \quad MR\sqrt{\frac{p_0}{N}} \log d \text{ when } \alpha \to 2, \quad \text{and} \quad MR\sqrt{\frac{p_0}{N}} \text{ when } \alpha > 2. \quad (4)$$

These results beg the question of tightness: are they improvable? As we see presently, they are not.

## 2.2 Algorithms for attaining the minimax rate

To show that the lower bounds of Proposition 1 and its subsequent specializations are sharp, we review a few stochastic gradient algorithms. We begin with stochastic gradient descent (SGD): SGD repeatedly samples $\xi \sim P$, computes $g \in \partial_x F(x; \xi)$, then performs the update $x \leftarrow \Pi_{\mathcal{X}}(x - \eta g)$, where $\eta$ is a stepsize parameter and $\Pi_{\mathcal{X}}$ denotes Euclidean projection onto $\mathcal{X}$. Standard analyses of stochastic gradient descent [10] show that after $N$ samples $\xi^i$, the SGD estimator $\widehat{x}(N)$ satisfies

$$\mathbb{E}[f(\widehat{x}(N))] - \inf_{x \in \mathcal{X}} f(x) \leq \mathcal{O}(1) \frac{R_2 M (\sum_{j=1}^d p_j)^{\frac{1}{2}}}{\sqrt{N}}, \quad (5)$$

where $R_2$ denotes the $\ell_2$-radius of $\mathcal{X}$. Dual averaging, due to Nesterov [11] (sometimes called "follow the regularized leader" [5]) is a more recent algorithm. In dual averaging, one again samples $g \in \partial_x F(x; \xi)$, but instead of updating the parameter vector $x$ one updates a dual vector $z$ by $z \leftarrow z + g$, then computes

$$x \leftarrow \operatorname*{argmin}_{x \in \mathcal{X}} \left\{ \langle z, x \rangle + \frac{1}{\eta} \psi(x) \right\},$$

where $\psi(x)$ is a strongly convex function defined over $\mathcal{X}$ (often one takes $\psi(x) = \frac{1}{2} \|x\|_2^2$). As we discuss presently, the dual averaging algorithm is somewhat more natural in asynchronous and parallel computing environments, and it enjoys the same type of convergence guarantees (5) as SGD.

The ADAGRAD algorithm [4, 9] is an extension of the preceding stochastic gradient methods. It maintains a diagonal matrix $S$, where upon receiving a new sample $\xi$, ADAGRAD performs the following: it computes $g \in \partial_x F(x; \xi)$, then updates

$$S_j \leftarrow S_j + g_j^2 \text{ for } j \in [d].$$

The dual averaging variant of ADAGRAD updates the usual dual vector $z \leftarrow z + g$; the update to $x$ is based on $S$ and a stepsize $\eta$ and computes

$$x \leftarrow \operatorname*{argmin}_{x' \in \mathcal{X}} \left\{ \langle z, x' \rangle + \frac{1}{2\eta} \left\langle x', S^{\frac{1}{2}} x' \right\rangle \right\}.$$

After $N$ samples $\xi$, the averaged parameter $\widehat{x}(N)$ returned by ADAGRAD satisfies

$$\mathbb{E}[f(\widehat{x}(N))] - \inf_{x \in \mathcal{X}} f(x) \leq \mathcal{O}(1)\frac{R_\infty M}{\sqrt{N}} \sum_{j=1}^{d} \sqrt{p_j}, \tag{6}$$

where $R_\infty$ denotes the $\ell_\infty$-radius of $\mathcal{X}$ (cf. [4, Section 1.3 and Theorem 5]). By inspection, the ADAGRAD rate (6) matches the lower bound in Proposition 1 and is thus optimal. It is interesting to note, though, that in the power law setting of Corollary 2 (recall the error order (4)), a calculation shows that the multiplier for the SGD guarantee (5) becomes $R_\infty \sqrt{d} \max\{d^{(1-\alpha)/2}, 1\}$, while ADAGRAD attains rate at worst $R_\infty \max\{d^{1-\alpha/2}, \log d\}$. For $\alpha > 1$, the ADAGRAD rate is no worse, and for $\alpha \geq 2$, is more than $\sqrt{d}/\log d$ better—an exponential improvement in the dimension.

## 3 Parallel and asynchronous optimization with sparsity

As we note in the introduction, recent works [12, 14] have suggested that sparsity can yield benefits in our ability to *parallelize* stochastic gradient-type algorithms. Given the optimality of ADAGRAD-type algorithms, it is natural to focus on their parallelization in the hope that we can leverage their ability to "adapt" to sparsity in the data. To provide the setting for our further algorithms, we first revisit Niu et al.'s HOGWILD! [12]. HOGWILD! is an asynchronous (parallelized) stochastic gradient algorithm for optimization over product-space domains, meaning that $\mathcal{X}$ in problem (1) decomposes as $\mathcal{X} = \mathcal{X}_1 \times \cdots \times \mathcal{X}_d$, where $\mathcal{X}_j \subset \mathbb{R}$. Fix a stepsize $\eta > 0$. A pool of independently running processors then performs the following updates asynchronously to a centralized vector $x$:

1. Sample $\xi \sim P$
2. Read $x$ and compute $g \in \partial_x F(x; \xi)$
3. For each $j$ s.t. $g_j \neq 0$, update $x_j \leftarrow \Pi_{\mathcal{X}_j}(x_j - \eta g_j)$.

Here $\Pi_{\mathcal{X}_j}$ denotes projection onto the $j$th coordinate of the domain $\mathcal{X}$. The key of HOGWILD! is that in step 2, the parameter $x$ is allowed to be inconsistent—it may have received partial gradient updates from many processors—and for appropriate problems, this inconsistency is negligible. Indeed, Niu et al. [12] show linear speedup in optimization time as the number of processors grow; they show this empirically in many scenarios, providing a proof under the somewhat restrictive assumptions that there is at most one non-zero entry in any gradient $g$ and that $f$ has Lipschitz gradients.

### 3.1 Asynchronous dual averaging

A weakness of HOGWILD! is that it appears only applicable to problems for which the domain $\mathcal{X}$ is a product space, and its analysis assumes $\|g\|_0 = 1$ for all gradients $g$. In effort to alleviate these difficulties, we now develop and present our asynchronous dual averaging algorithm, ASYNCDA. ASYNCDA maintains and upates a centralized dual vector $z$ instead of a parameter $x$, and a pool of processors perform asynchronous updates to $z$, where each processor independently iterates:

1. Read $z$ and compute $x := \operatorname{argmin}_{x \in \mathcal{X}} \left\{ \langle z, x \rangle + \frac{1}{\eta}\psi(x) \right\}$     // *Implicitly increment "time" counter $t$ and let $x(t) = x$*
2. Sample $\xi \sim P$ and let $g \in \partial_x F(x; \xi)$   // *Let $g(t) = g$.*
3. For $j \in [d]$ such that $g_j \neq 0$, update $z_j \leftarrow z_j + g_j$.

Because the actual computation of the vector $x$ in ASYNCDA is performed locally on each processor in step 1 of the algorithm, the algorithm can be *executed* with any proximal function $\psi$ and domain $\mathcal{X}$. The only communication point between any of the processors is the addition operation in step 3. Since addition is commutative and associative, forcing all asynchrony to this point of the algorithm is a natural strategy for avoiding synchronization problems.

In our analysis of ASYNCDA, and in our subsequent analysis of the adaptive methods, we require a measurement of time elapsed. With that in mind, we let $t$ denote a time index that exists (roughly) behind-the-scenes. We let $x(t)$ denote the vector $x \in \mathcal{X}$ computed in the $t$th step 1 of the ASYNCDA

algorithm, that is, whichever is the $t$th $x$ actually computed by any of the processors. This quantity exists and is recoverable from the algorithm, and it is possible to track the running sum $\sum_{\tau=1}^{t} x(\tau)$.

Additionally, we state two assumptions encapsulating the conditions underlying our analysis.

**Assumption A.** *There is an upper bound $m$ on the delay of any processor. In addition, for each $j \in [d]$ there is a constant $p_j \in [0,1]$ such that $P(\xi_j \neq 0) \leq p_j$.*

We also require certain continuity (Lipschitzian) properties of the loss functions; these amount to a second moment constraint on the instantaneous $\partial F$ and a rough measure of gradient sparsity.

**Assumption B.** *There exist constants $\mathsf{M}$ and $(M_j)_{j=1}^{d}$ such that the following bounds hold for all $x \in \mathcal{X}$: $\mathbb{E}[\|\partial_x F(x;\xi)\|_2^2] \leq \mathsf{M}^2$ and for each $j \in [d]$ we have $\mathbb{E}[|\partial_{x_j} F(x;\xi)|] \leq p_j M_j$.*

With these definitions, we have the following theorem, which captures the convergence behavior of ASYNCDA under the assumption that $\mathcal{X}$ is a Cartesian product, meaning that $\mathcal{X} = \mathcal{X}_1 \times \cdots \times \mathcal{X}_d$, where $\mathcal{X}_j \subset \mathbb{R}$, and that $\psi(x) = \frac{1}{2}\|x\|_2^2$. Note the algorithm itself can still be efficiently parallelized for more general convex $\mathcal{X}$, even if the theorem does not apply.

**Theorem 3.** *Let Assumptions A and B and the conditions in the preceding paragraph hold. Then*

$$\mathbb{E}\left[\sum_{t=1}^{T} F(x(t);\xi^t) - F(x^*;\xi^t)\right] \leq \frac{1}{2\eta}\|x^*\|_2^2 + \frac{\eta}{2}T\mathsf{M}^2 + \eta Tm\sum_{j=1}^{d} p_j^2 M_j^2.$$

We now provide a few remarks to explain and simplify the result. Under the more stringent condition that $|\partial_{x_j} F(x;\xi)| \leq M_j$, Assumption A implies $\mathbb{E}[\|\partial_x F(x;\xi)\|_2^2] \leq \sum_{j=1}^{d} p_j M_j^2$. Thus, for the remainder of this section we take $\mathsf{M}^2 = \sum_{j=1}^{d} p_j M_j^2$, which upper bounds the Lipschitz continuity constant of the objective function $f$. We then obtain the following corollary.

**Corollary 4.** *Define $\widehat{x}(T) = \frac{1}{T}\sum_{t=1}^{T} x(t)$, and set $\eta = \|x^*\|_2 / \mathsf{M}\sqrt{T}$. Then*

$$\mathbb{E}[f(\widehat{x}(T)) - f(x^*)] \leq \frac{\mathsf{M}\|x^*\|_2}{\sqrt{T}} + m\frac{\|x^*\|_2}{2\mathsf{M}\sqrt{T}}\sum_{j=1}^{d} p_j^2 M_j^2.$$

Corollary 4 is nearly immediate: since $\xi^t$ is independent of $x(t)$, we have $\mathbb{E}[F(x(t);\xi^t) \mid x(t)] = f(x(t))$; applying Jensen's inequality to $f(\widehat{x}(T))$ and performing an algebraic manipulation give the result. If the data is suitably sparse, meaning that $p_j \leq 1/m$, the bound in Corollary 4 simplifies to

$$\mathbb{E}[f(\widehat{x}(T)) - f(x^*)] \leq \frac{3}{2}\frac{\mathsf{M}\|x^*\|_2}{\sqrt{T}} = \frac{3}{2}\frac{\sqrt{\sum_{j=1}^{d} p_j M_j^2}\,\|x^*\|_2}{\sqrt{T}}, \tag{7}$$

which is the convergence rate of stochastic gradient descent even in centralized settings (5). The convergence guarantee (7) shows that after $T$ timesteps, the error scales as $1/\sqrt{T}$; however, if we have $k$ processors, updates occur roughly $k$ times as quickly, as they are asynchronous, and in time scaling as $N/k$, we can evaluate $N$ gradient samples: a linear speedup.

### 3.2 Asynchronous AdaGrad

We now turn to extending ADAGRAD to asynchronous settings, developing ASYNCADAGRAD (asynchronous ADAGRAD). As in the ASYNCDA algorithm, ASYNCADAGRAD maintains a shared dual vector $z$ (the sum of gradients) and the shared matrix $S$, which is the diagonal sum of squares of gradient entries (recall Section 2.2). The matrix $S$ is initialized as $\mathrm{diag}(\delta^2)$, where $\delta_j \geq 0$ is an initial value. Each processor asynchronously performs the following iterations:

1. Read $S$ and $z$ and set $G = S^{\frac{1}{2}}$. Compute $x := \operatorname{argmin}_{x \in \mathcal{X}}\{\langle z, x\rangle + \frac{1}{2\eta}\langle x, Gx\rangle\}$   *// Implicitly increment "time" counter $t$ and let $x(t) = x$, $S(t) = S$*

2. Sample $\xi \sim P$ and let $g \in \partial F(x;\xi)$

3. For $j \in [d]$ such that $g_j \neq 0$, update $S_j \leftarrow S_j + g_j^2$ and $z_j \leftarrow z_j + g_j$.

As in the description of ASYNCDA, we note that $x(t)$ is the vector $x \in \mathcal{X}$ computed in the $t$th "step" of the algorithm (step 1), and similarly associate $\xi^t$ with $x(t)$.

To analyze ASYNCADAGRAD, we make a somewhat stronger assumption on the sparsity properties of the losses $F$ than Assumption B.

**Assumption C.** *There exist constants $(M_j)_{j=1}^d$ such that $\mathbb{E}[(\partial_{x_j} F(x;\xi))^2 \mid \xi_j \neq 0] \leq M_j^2$ for all $x \in \mathcal{X}$.*

Indeed, taking $\mathsf{M}^2 = \sum_j p_j M_j^2$ shows that Assumption C implies Assumption B with specific constants. We then have the following convergence result.

**Theorem 5.** *In addition to the conditions of Theorem 3, let Assumption C hold. Assume that for all $j$ we have $\delta^2 \geq M_j^2 m$ and $\mathcal{X} \subset [-R_\infty, R_\infty]^d$. Then*

$$\sum_{t=1}^T \mathbb{E}\left[F(x(t);\xi^t) - F(x^*;\xi^t)\right]$$

$$\leq \sum_{j=1}^d \min\left\{\frac{1}{\eta}R_\infty^2 \mathbb{E}\left[\left(\delta^2 + \sum_{t=1}^T g_j(t)^2\right)^{\frac{1}{2}}\right] + \eta\mathbb{E}\left[\left(\sum_{t=1}^T g_j(t)^2\right)^{\frac{1}{2}}\right](1 + p_j m), M_j R_\infty p_j T\right\}.$$

It is possible to relax the condition on the initial constant diagonal term; we defer this to the full version of the paper.

It is natural to ask in which situations the bound provided by Theorem 5 is optimal. We note that, as in the case with Theorem 3, we may obtain a convergence rate for $f(\widehat{x}(T)) - f(x^*)$ using convexity, where $\widehat{x}(T) = \frac{1}{T}\sum_{t=1}^T x(t)$. By Jensen's inequality, we have for any $\delta$ that

$$\mathbb{E}\left[\left(\delta^2 + \sum_{t=1}^T g_j(t)^2\right)^{\frac{1}{2}}\right] \leq \left(\delta^2 + \sum_{t=1}^T \mathbb{E}[g_j(t)^2]\right)^{\frac{1}{2}} \leq \sqrt{\delta^2 + Tp_j M_j^2}.$$

For interpretation, let us now make a few assumptions on the probabilities $p_j$. If we assume that $p_j \leq c/m$ for a universal (numerical) constant $c$, then Theorem 5 guarantees that

$$\mathbb{E}[f(\widehat{x}(T)) - f(x^*)] \leq \mathcal{O}(1)\left[\frac{1}{\eta}R_\infty^2 + \eta\right]\sum_{j=1}^d M_j \min\left\{\frac{\sqrt{\log(T)/T + p_j}}{\sqrt{T}}, p_j\right\}, \quad (8)$$

which is the convergence rate of ADAGRAD except for a small factor of $\min\{\sqrt{\log T/T}, p_j\}$ in addition to the usual $\sqrt{p_j/T}$ rate. In particular, optimizing by choosing $\eta = R_\infty$, and assuming $p_j \gtrsim \frac{1}{T}\log T$, we have convergence guarantee

$$\mathbb{E}[f(\widehat{x}(T)) - f(x^*)] \leq \mathcal{O}(1)R_\infty \sum_{j=1}^d M_j \min\left\{\frac{\sqrt{p_j}}{\sqrt{T}}, p_j\right\},$$

which is minimax optimal by Proposition 1.

In fact, however, the bounds of Theorem 5 are somewhat stronger: they provide bounds using the *expectation* of the squared gradients $g_j(t)$ rather than the maximal value $M_j$, though the bounds are perhaps clearer in the form (8). We note also that our analysis applies to more adversarial settings than stochastic optimization (e.g., to online convex optimization [5]). Specifically, an adversary may choose an arbitrary sequence of functions subject to the random data sparsity constraint (2), and our results provide an expected regret bound, which is strictly stronger than the stochastic convergence guarantees provided (and guarantees high-probability convergence in stochastic settings [3]). Moreover, our comments in Section 2 about the relative optimality of ADAGRAD versus standard gradient methods apply. When the data is sparse, we indeed should use asynchronous algorithms, but using adaptive methods yields even more improvement than simple gradient-based methods.

## 4 Experiments

In this section, we give experimental validation of our theoretical results on ASYNCADAGRAD and ASYNCDA, giving results on two datasets selected for their high-dimensional sparsity.[2]

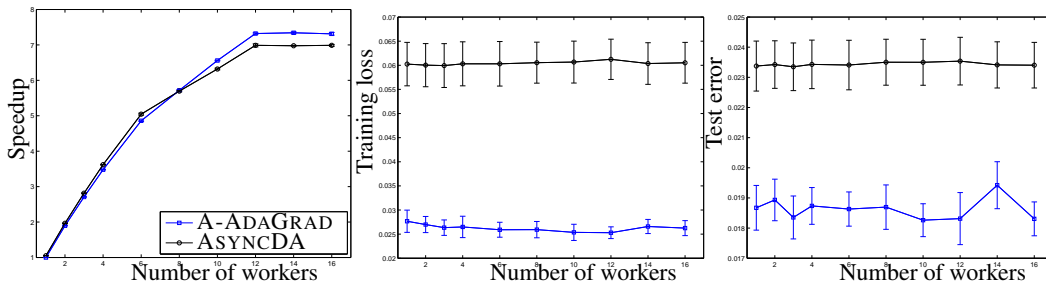

**Figure 1.** Experiments with URL data. Left: speedup relative to one processor. Middle: training dataset loss versus number of processors. Right: test set error rate versus number of processors. A-ADAGRAD abbreviates ASYNCADAGRAD.

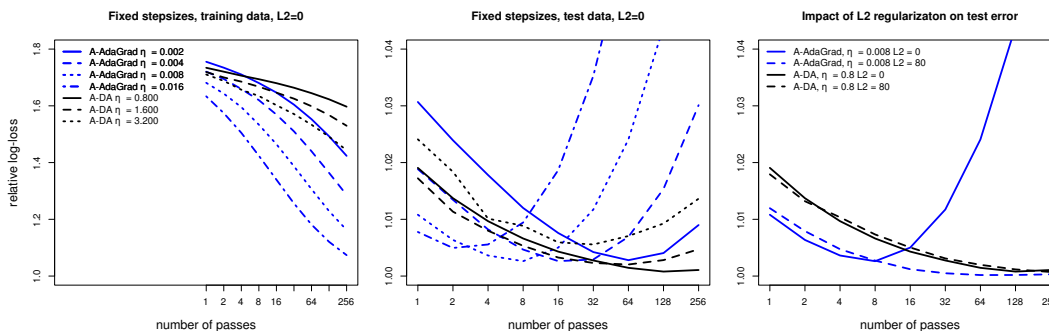

**Figure 2:** Relative accuracy for various stepsize choices on an click-through rate prediction dataset.

## 4.1 Malicious URL detection

For our first set of experiments, we consider the speedup attainable by applying ASYNCADAGRAD and ASYNCDA, investigating the performance of each algorithm on a malicious URL prediction task [7]. The dataset in this case consists of an anonymized collection of URLs labeled as malicious (e.g., spam, phishing, etc.) or benign over a span of 120 days. The data in this case consists of $2.4 \cdot 10^6$ examples with dimension $d = 3.2 \cdot 10^6$ (sparse) features. We perform several experiments, randomly dividing the dataset into $1.2 \cdot 10^6$ training and test samples for each experiment.

In Figure 1 we compare the performance of ASYNCADAGRAD and ASYNCDA after doing after single pass through the training dataset. (For each algorithm, we choose the stepsize $\eta$ for optimal training set performance.) We perform the experiments on a single machine running Ubuntu Linux with six cores (with two-way hyperthreading) and 32Gb of RAM. From the left-most plot in Fig. 1, we see that up to six processors, both ASYNCDA and ASYNCADAGRAD enjoy the expected linear speedup, and from 6 to 12, they continue to enjoy a speedup that is linear in the number of processors though at a lesser slope (this is the effect of hyperthreading). For more than 12 processors, there is no further benefit to parallelism on this machine.

The two right plots in Figure 1 plot performance of the different methods (with standard errors) versus the number of worker threads used. Both are essentially flat; increasing the amount of parallelism does nothing to the average training loss or the test error rate for either method. It is clear, however, that for this dataset, the adaptive ASYNCADAGRAD algorithm provides substantial performance benefits over ASYNCDA.

## 4.2 Click-through-rate prediction experiments

We also experiment on a proprietary datasets consisting of search ad impressions. Each example corresponds to showing a search-engine user a particular text ad in response to a query string. From this, we construct a very sparse feature vector based on the text of the ad displayed and the query string (no user-specific data is used). The target label is 1 if the user clicked the ad and -1 otherwise.

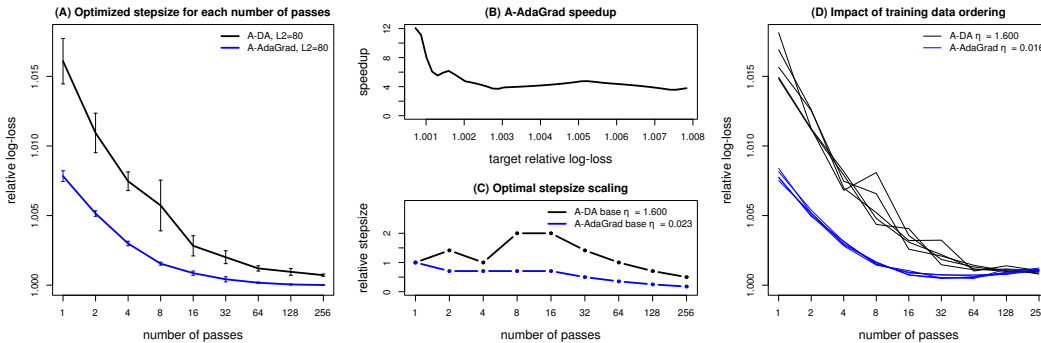

**Figure 3.** (A) Relative test-set log-loss for ASYNCDA and ASYNCADAGRAD, choosing the best stepsize (within a factor of about $1.4\times$) individually for each number of passes. (B) Effective speedup for ASYNCADAGRAD. (C) The best stepsize $\eta$, expressed as a scaling factor on the stepsize used for one pass. (D) Five runs with different random seeds for each algorithm (with $\ell_2$ penalty 80).

We fit logistic regression models using both ASYNCDA and ASYNCADAGRAD. We run extensive experiments on a moderate-sized dataset (about $10^7$ examples, split between training and testing),[3] which allows thorough investigation of the impact of the stepsize $\eta$, the number of training passes,[3] and $\ell_2$-regularization on accuracy. For these experiments we used 32 threads on 16 core machines for each run, as ASYNCADAGRAD and ASYNCDA achieve similar speedups from parallelization.

On this dataset, ASYNCADAGRAD typically achieves an effective *additional* speedup over ASYNCDA of $4\times$ or more. That is, to reach a given level of accuracy, ASYNCDA generally needs four times as many effective passes over the dataset. We measure accuracy with log-loss (the logistic loss) averaged over five runs using different random seeds (which control the order in which the algorithms sample examples during training). We report relative values in Figures 2 and 3, that is, the ratio of the mean loss for the given datapoint to the lowest (best) mean loss obtained. Our results are not particularly sensitive to the choice of relative log-loss as the metric of interest; we also considered AUC (the area under the ROC curve) and observed similar results.

Figure 2 shows relative log-loss as a function of the number of training passes for various stepsizes. Without regularization, ASYNCADAGRAD is prone to overfitting: it achieves significantly higher accuracy on the training data (Fig. 2 (left)), but unless the stepsize is tuned carefully to the number of passes, it will overfit (Fig. 2 (middle)). Fortunately, the addition of $\ell_2$ regularization largely solves this problem. Indeed, Figure 2 (right) shows that while adding an $\ell_2$ penalty of 80 has very little impact on ASYNCDA, it effectively prevents the overfitting of ASYNCADAGRAD.[4]

Fixing $\ell_2$ regularization multiplier to 80, we varied the stepsize $\eta$ over a multiplicative grid with resolution $\sqrt{2}$ for each number of passes and for each algorithm. Figure 3 reports the results obtained by selecting the best stepsize in terms of test set log-loss for each number of passes. Figure 3(A) shows relative log-loss of the best stepsize for each algorithm; 3(B) shows the relative time ASYNCDA requires with respect to ASYNCADAGRAD to achieve a given loss. Specifically, Fig. 3(B) shows the ratio of the number of passes the algorithms require to achieve a fixed loss, which gives a broader estimate of the speedup obtained by using ASYNCADAGRAD; speedups range from $3.6\times$ to $12\times$. Figure 3(C) shows the optimal stepsizes as a function of the best setting for one pass. The optimal stepsize decreases moderately for ASYNCADAGRAD, but are somewhat noisy for ASYNCDA.

It is interesting to note that ASYNCADAGRAD's accuracy is largely independent of the ordering of the training data, while ASYNCDA shows significant variability. This can be seen both in the error bars on Figure 3(A), and explicitly in Figure 3(D), where we plot one line for each of the five random seeds used. Thus, while on the one hand ASYNCDA requires somewhat less tuning of the stepsize and $\ell_2$ parameter, tuning ASYNCADAGRAD is much easier because of its predictable response.

## Footnotes

[1] Formally, if $\pi_\xi$ denotes the coordinate projection zeroing all indices $j$ of its argument where $\xi_j = 0$, then $F(\pi_\xi(x); \xi) = F(x; \xi)$ for all $x, \xi$. This follows from the first-order conditions for convexity [6].

[2] In our experiments, ASYNCDA and HOGWILD! had effectively identical performance.

[3]Here "number of passes" more precisely means the expected number of times each example in the dataset is trained on. That is, each worker thread randomly selects a training example from the dataset for each update, and we continued making updates until (dataset size) $\times$ (number of passes) updates have been processed.

[4]For both algorithms, this is accomplished by adding the term $\eta 80 \|x\|_2^2$ to the $\psi$ function. We can achieve slightly better results for ASYNCADAGRAD by varying the $\ell_2$ penalty with the number of passes.

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
