[Reviews · NeurIPS 2013]

Submitted by Assigned_Reviewer_4

The authors derive lower bounds on the optimization error for solving a class of stochastic optimization problem with sparsity conditions. A parallel and asynchronous algorithm called PROSCIUTTO is developed and analyzed. Numerical comparisons with HOGWILD on Malicious URL detection and Click-through-rate prediction experiments are provided.


Quality: The paper is technically sound. The proofs are provided in supplementary material.

Clarity: The paper is well-organized.

Originality: A parallel and asynchronous algorithm for high-dimensional statistical learning sounds interesting.


Significance: This paper provides some understanding on the fundamental limits of optimization and learning algorithms when the data is sparse.
Summary: The fundamental limits of optimization algorithms is interesting. A parallel and asynchronous algorithm for high-dimensional statistical learning is useful.

Submitted by Assigned_Reviewer_5

The paper presents a discussion of (supervised linear) learning in the presence of sparse features. It analyzes convergence of common approaches like SGD, AdaGrad based on a best-case analysis. Based on that, it proposes a variant of AdaGrad for distributed optimization which is inspired by HogWild. Experiments show a speedup (in terms of the number of dataset passes) over the latter.

The paper is reasonably well written. In some places, it would be nice to have some more intuitions provided for the analytical results, though. In particular, Section 2 and the derived bounds would benefit greatly from this. For example, just provide the relationship between various typical sparsity levels (english text, web pages, ...) and the best case convergence.

Some comments about the experiments: It would be nice to also see convergence plots for different numbers of CPUs in wall clock time. This would facilitate some insight into the trade-offs between (potentially) conflicting updates and the speedup provided by more cores in different parts of the optimization.

Also, in Section 4.2 the train/test split is suggested to be "even". Does this mean random or by time?
Summary: Good paper that could benefit from some more intuitions and experimental updates for the camera ready version.

Submitted by Assigned_Reviewer_6

This paper studies the effect of data sparsity on learning and optimization. Specifically, the paper assumes that the support of gradient of an objective function is in the support of feature vectors, where feature vectors are considered to be sparse. Definding appearance probabablities on feature, the paper derives a lower bound on the gap between an estimated solution from minimization with given data and a true minimizer, and shows that ADAGRAD methods [4,8] achieves this lower bound (and thereby optimal). The paper also suggests an algorithm using asynchronous updates, showing that the suggested algorithm achieves the optimal bound up to some factors depending on log(dimensionality) and update delay probabilities.

Quality:
The claims are supported by proved theorems. Although the claims are technically sound, I believe the main assumption in Eq.(2) excludes many popular approaches with regularization, for example L1 (lasso), L2 (ridge), group lasso, etc. The weakness is indeed revealed in the experiments, for instance the right-hand side plots in Figure 2. The plots shows the sensitivity of the suggested algorithm to the choices of the steplength. In the experiments the authors chose "optimal" steplengths by using test sets, but this doesn't sound very practical.

Also, it is assumed that the distance from independence is bounded by some constant m, that is, di(\rho) <= m, but I believe, depending on systems, it could be proportional to the number of parallel workers k. If it is the case, then the linear speedup using multiple cores in the end of Section 3.2 is unlikely. The machines used in the experiment had only a limited number of cores, so the validity of assumption was not very well supported.

Clarity:
In general the arguments are clearly stated and well organized. But some minor points can be improved for better readability:
- page 2, line 104: consist distributions -> consist of distributions
- page 3, Eq (6) and (7): It would be kind to note the fact that R_2 ~ O(\sqrt(d)) for less-trained eyes.
- page 3, the last line: exponential improvement in the dimension: it is not clear to me why "exponential" in the dimension.
- page 5, line 236: Assuming the data appearance... -> Assuming the feature appearance

Originality:
I believe the introduction of the assumption in Eq.(2), the lower bound in Proposition 1, the introduction of feature appearance probability in this context, are new. The analysis for Hogwild algorithm seems to be new as well, but I didn't go through the proof of that part.

Significance:
The new analysis supporting asynchronous update when features are sparse, will be an interesting message to the community. However as the assumption in Eq.(2) excludes many interesting problems, I don't think it will have a large impact.
Summary: The paper suggests an analysis showing that the optimal bounds can be achieved by their suggested algorithm up to some factors, where the algorithm is claimed to have linear speedup with the number of cores and to benefit from the sparsity of input features. However, its applicability seems to be limited due to a rather restrictive assumption it is based upon, which makes the suggested method sensitive to the choices of stepsizes, and the other assumption on asynchronous update delays.
Author Feedback

Author rebuttal: We thank the reviewers for their feedback and comments on the paper, all of
which we plan to address in the final version of the paper. We address major
comments in turn.

REVIEWER 4

We agree with the reviewer's feedback.

REVIEWER 5

We agree with the reviewer's suggestion to give more intuition and
examples. In particular we have in mind making use of the following
reference: http://arxiv.org/abs/0706.1062, which provides some telling
phenomenology that motivate a focus on data sparseness.

We tried to plot speedups in experiments as we found them to be the most
easily interpretable; the speedups are measured in actual wall-clock time.

In Sec 4.2, we used a random half of the data for training and the
other half for testing; we will clarify the wording.

We have been actively working on new experiments since submission; see
also our responses to reviewer 6.

REVIEWER 6

The reviewer is correct that Eq.(2) excludes applying a regularization penalty
on the full coefficient vector for each training example; however, in practice
it is not difficult to circumvent this restriction using tricks that are
natural when working in the large dataset setting. E.g.,

(A) Rather than putting some regularization penalty on each example, one could
make use of essentially regularization-only examples which are sampled with
probability 1/N for N large, potentially N=(# training examples). This
would not hurt sparsity.

(B) Pre-process the dataset to count the number of occurrences of each feature,
say k. Then, to obtain a total L2 penalty of lambda for the whole dataset,
apply an L2 penalty of lambda/k to the coefficient for the feature each
time it occurs in a sampled training example. This ensures the gradient with
L2 penalty has the same support as the underlying loss, and in expectation the
L2 penalty is correct.

In terms of the assumption on di(\rho) \le m, the reviewer is right that m
could be as large as k in the worst case. In practice, we do not see a penalty
from this; indeed, in the intervening time between submission and reviews, we
have also simplified the analysis and removed this dependence (roughly, the
dependence is removed with the introduction of a p_j di(\rho) term instead of
\sqrt{p_j} di(\rho) term, the former of which is always stronger).

We will address all clarity concerns the reviewer notes--thanks for suggesting
these.

The reviewer suggests that the method has limited applicability based on our
sparsity assumptions. This is true. At the same time, many problems do have
sparse data. Moreover, in practice we have not seen such limited
applicability; in additional experiments with image and speech data we see
similar benefits to those reported in the paper. We are also actively
experimenting with other datasets and machine sizes to find the limitations of
the method; we plan to update our experimental work to reflect these. (As the
reviewer notes, it is interesting to see how many processors are possible
while still enjoying linear or nearly linear speedups, as there is presumably
a breaking point; we have seen linear speedups through 12 processors and are
experimenting currently with larger machines. Additionally, using the data
from Fig. (2), we simulated a constant update delay of K (e.g., K cores) for
one pass over the training set, and we do not see a significant drop in test
set or training set AUC loss for K < 1000 cores; above that, performance
degrades. We will include these experiments in the final version. Note that we
do not measure timing for these, as we are simulating only to evaluate
losses.)